# Pharmacological Evidence on Augmented Antiallodynia Following Systemic Co-Treatment with GlyT-1 and GlyT-2 Inhibitors in Rat Neuropathic Pain Model

**DOI:** 10.3390/ijms22052479

**Published:** 2021-03-01

**Authors:** Amir Mohammadzadeh, Péter P. Lakatos, Mihály Balogh, Ferenc Zádor, Dávid Árpád Karádi, Zoltán S. Zádori, Kornél Király, Anna Rita Galambos, Szilvia Barsi, Pál Riba, Sándor Benyhe, László Köles, Tamás Tábi, Éva Szökő, Laszlo G. Harsing, Mahmoud Al-Khrasani

**Affiliations:** 1Department of Pharmacology and Pharmacotherapy, Faculty of Medicine, Semmelweis University, Nagyvárad tér 4, P.O. Box 370, H-1445 Budapest, Hungary; mohammadzadeh.amir@med.semmelweis-univ.hu (A.M.); balogh.mihaly@med.semmelweis-univ.hu (M.B.); zador.ferenc@pharma.semmelweis-univ.hu (F.Z.); karadi.david_arpad@med.semmelweis-univ.hu (D.Á.K.); zadori.zoltan@med.semmelweis-univ.hu (Z.S.Z.); kiraly.kornel@med.semmelweis-univ.hu (K.K.); galambos.anna@pharma.semmelweis-univ.hu (A.R.G.); barsi.szilvia@pharma.semmelweis-univ.hu (S.B.); riba.pal@med.semmelweis-univ.hu (P.R.); koles.laszlo@med.semmelweis-univ.hu (L.K.); harsing.laszlo@med.semmelweis-univ.hu (L.G.H.J.); 2Department of Pharmacodynamics, Faculty of Pharmacy, Semmelweis University, Nagyvárad tér 4, H-1089 Budapest, Hungary; lakatos.peter@pharma.semmelweis-univ.hu (P.P.L.); tabi.tamas@pharma.semmelweis-univ.hu (T.T.); szoko.eva@pharma.semmelweis-univ.hu (É.S.); 3Institute of Biochemistry, Biological Research Center, Temesvári krt. 62, H-6726 Szeged, Hungary; benyhe.sandor@brc.hu

**Keywords:** neuropathic pain, NFPS, Org-25543, CSF glycine content, glycine transporter inhibitor combination

## Abstract

The limited effect of current medications on neuropathic pain (NP) has initiated large efforts to develop effective treatments. Animal studies showed that glycine transporter (GlyT) inhibitors are promising analgesics in NP, though concerns regarding adverse effects were raised. We aimed to study NFPS and Org-25543, GlyT-1 and GlyT-2 inhibitors, respectively and their combination in rat mononeuropathic pain evoked by partial sciatic nerve ligation. Cerebrospinal fluid (CSF) glycine content was also determined by capillary electrophoresis. Subcutaneous (s.c.) 4 mg/kg NFPS or Org-25543 showed analgesia following acute administration (30–60 min). Small doses of each compound failed to produce antiallodynia up to 180 min after the acute administration. However, NFPS (1 mg/kg) produced antiallodynia after four days of treatment. Co-treatment with subanalgesic doses of NFPS (1 mg/kg) and Org-25543 (2 mg/kg) produced analgesia at 60 min and thereafter meanwhile increased significantly the CSF glycine content. This combination alleviated NP without affecting motor function. Test compounds failed to activate G-proteins in spinal cord. To the best of our knowledge for the first time we demonstrated augmented analgesia by combining GlyT-1 and 2 inhibitors. Increased CSF glycine content supports involvement of glycinergic system. Combining selective GlyT inhibitors or developing non-selective GlyT inhibitors might have therapeutic value in NP.

## 1. Introduction

Despite the fact that a lot of analgesic drugs are increasingly prescribed either alone or in combination for the treatment of neuropathic pain, satisfactory treatment has not been achieved yet. 

Currently, several classes of drugs are used for the management of neuropathic pain (NP). First line drugs include tricyclic antidepressants, dual reuptake inhibitors of serotonin and norepinephrine, gabapentinoids and 5% lidocaine transdermal patch. The second line drugs are classical opioids and drugs having mixed opioid and non-opioid actions, like tramadol. Third line drugs include antiepileptics, topical capsaicin, memantine and mexiletine [1,2]. The molecular targets of the current medications include biogenic amine transporters, voltage-gated N-type calcium and sodium channels, opioid receptors, N-methyl-D-aspartate (NMDA) receptor and transient receptor potential vanilloid subtype 1 (TRPV1) [1,2,3]. These targets do influence the pain transduction at the level of the central nervous system (CNS) and peripheral traffic points. The spinal dorsal horn is a crucial point in conveying pain sensation from the periphery to the brain [4].

Growing body of evidence supports the efficacy of glycine transporter (GlyT) inhibitors in the attenuation of NP in animal pain models [5,6,7,8]. The antiallodynic actions of the inhibitors of both GlyTs, namely neural/astrocytic type-1 (GlyT-1) and the neuronal type-2 (GlyT-2), have been demonstrated in neuropathic conditions evoked by partial sciatic nerve ligation (pSNL) in mice or rats following spinal administration [9,10,11]. Orally or intravenously administered sarcosine (GlyT-1 selective substrate inhibitor) showed the antiallodynic effect, though the onset of drug action was conspicuously inconsistent [9,11]. On the other hand, the time lag between systemic administration of GlyT-2 selective inhibitors and the appearance of analgesic effects was short.

So far, studies found that injury of peripheral sensory neurons or inflammation results in reduced function of glycinergic interneurons in the spinal dorsal horn and are manifested by development of neuropathic pain [5,12]. Restoring these neurochemical changes by GlyT-1 or GlyT-2 inhibitors might be of great importance in attenuating NP. We have recently hypothesized that the combination of the GlyT inhibitors might offer augmented analgesia [5].

Noteworthy, concerns have also been raised about the adverse effects of GlyT inhibitors, though doses that cause these adverse effects are phenotype (rat or mice) dependent [13,14].

Despite thorough studies on the analgesic action of GlyT inhibitors in neuropathic animal pain models, to the best of our knowledge no study had been carried out to test the antiallodynic effect of the combination of GlyT-1 and GlyT-2 selective inhibitors.

Based on currently available data, our aim was to assess the antiallodynic actions of subcutaneously administered NFPS (GlyT-1 selective irreversible inhibitor, Figure 1), after acute or chronic treatment, Org-25543 (GlyT-2 selective irreversible inhibitor, Figure 1) after acute treatment and the combination of NFPS and Org-25543 after acute treatment in rats with mononeuropathic pain evoked by pSNL. In this study, we also investigated the effects of GlyT-1 and GlyT-2 selective inhibitors either alone or in combination on rat motor coordination and balance by the rotarod test and the content of glycine level of the cerebrospinal fluid (CSF) and of spinal L4-L6 tissues. Finally, to exclude agonistic activity of NFPS or Org-25543 to G-protein coupled receptors, functional [^35^S]GTPγS binding assay was carried out in rat spinal cord membranes.

## 2. Results

### 2.1. Acute and Chronic Treatment with GlyT-1 Inhibitor NFPS

The GlyT-1 inhibitor, NFPS in doses of 1 and 2 mg/kg (s.c.) failed to show any significant antiallodynic effect assessed by paw pressure thresholds (PWT) after acute, systemic administration. On the other hand in 4 mg/kg dose showed significant antiallodynic effect 30 and 60 min after acute treatment compared to vehicle (30’: 31.34 ± 4.02 g vs. 18.55 ± 1.97 g, *n* = 5; 60’: 29.55 ± 2.93 g vs. 16.96 ± 2.88 g, *n* = 5, F(4, 140) = 2.518, *p* < 0.001, two-way ANOVA, Newman–Keuls post hoc test; Figure 2A). Chronic treatment with NFPS (1 mg/kg, s.c.) produced significant antiallodynic effect only at day 4 compared to the vehicle treated group (25.67 ± 0.88 g vs. 16.15 ± 1.18 g, *n* = 5, F(2, 60) = 0.5887, *p* < 0.01, two-way ANOVA, Newman–Keuls post hoc test; Figure 2B).

### 2.2. A Cute and Chronic Treatment with GlyT-2 Inhibitor Org-25543

The Org-25543 and GlyT-2 inhibitor was tested in doses of 2 and 4 mg/kg for its analgesic effect in pSNL evoked allodynia. Only a dose of 4 mg/kg s.c. showed an antiallodynic effect compared to the vehicle (25.92 ± 1.95 g vs. 18.22 ± 1.07 g, Org: *n* = 5 and vehicle: *n* = 6, F(4, 130) = 1.644, *p* < 0.01, two-way ANOVA, Newman–Keuls post hoc test; Figure 3A). Treatment with vehicle failed to show an impact on the operated paw. Daily treatment with 2 mg/kg Org-25543 for 4 days did not show any antiallodynic effect (Figure 3B).

### 2.3. Augmented Antiallodynic Effect by the Combination of the NFPS and Org-25543 in Neuropathic Rats

Acute treatment with NFPS (1 mg/kg, s.c.) or Org-25543 (2 mg/kg, s.c.) failed to produce antiallodynic effects. On the other hand, the combination of GlyT inhibitors in these doses produced antiallodynic effects 60 and 180 min post treatment compared to vehicle (60’: 25.6 ± 3.69 g vs. 15.56 ± 2.37 g; 180’: 29.56 ± 15.12 g, combination: *n* = 5, vehicle: *n* = 4, F(4, 130) = 0.1225, *p* < 0.01 and *p* < 0.001, respectively, two-way ANOVA, Newman–Keuls post hoc test; Figure 4). The vehicle applied to dissolve the GlyT inhibitors did not affect the thresholds of animals.

### 2.4. The Impact of NFPS and Org-25543 on Motor Function in Rats

NFPS (2–4 mg/kg, s.c.) and Org-25543 (4mg/kg, s.c.) failed to exhibit motor dysfunction in rats (Figure 5). In addition, the combination of the two drugs in the doses of 1 mg/kg for NFPS and 2 mg/kg for Org-25543 also failed to influence the rat motor function (Figure 5). Morphine treatment (6.4 mg/kg, s.c., positive control) resulted in disturbances in motor function manifested by shorter fall-off times in rats (47.29 ± 12.66 s vs. 178 ± 0.4 s, morphine: *n* = 7, vehicle: *n* = 5; F(2, 9) = 0.12, *p* < 0.001, one-way ANOVA, Newman–Keuls post-hoc test). In each treatment group 4–7 animals were used.).

### 2.5. Alteration of Spinal Cord Glycine Content

Partial sciatic nerve ligation in the vehicle group caused a moderate, non-significant elevation of CSF glycine concentration compared to the sham group (135.7% ± 15.24% vs. 100% ± 11.55%, Figure 6A). Either s.c. NFPS (1 mg/kg) or Org-25543 (2 mg/kg) also produced moderate non-significant elevation of CSF glycine content (156.3% ± 27.08% and 130.7% ± 56.10%, respectively, Figure 6A). The combination of the two compounds in the same doses caused a more pronounced, significant increase in the CSF glycine content compared to sham group (195.9% ± 35.32% vs. 100% ± 11.55%, combination: *n* = 9, sham: *n* = 11; F = 2.722, *p* < 0.01, one-way ANOVA, Holm–Sidak post-hoc test; Figure 6A). L-glutamate levels was significantly increased in CSF of operated animals compared to sham group, but was not modified by the treatment (159.3% ± 23.28% and 158.4% ± 9.21% vs. 100% ± 16.08%, vehicle: *n* = 4, combination: *n* = 5, sham: *n* = 7, F(2, 13) = 4.639, *p* < 0.05, one-way ANOVA, Holm–Sidak post-hoc test; Figure 6B). On the other hand, no significant changes were detected in the tissue content of glycine or L-glutamate of L4-L6 spinal segments in either ipsi- or contralateral side in rats undergone pSNL (Figure 6C,D). 

### 2.6. Lack of Agonist Activity of NFPS and Org-25543 in G-Protein Activity Assay

Figure 7 depicts the effect of NFPS and Org-25543 on [^35^S]GTPγS binding in spinal tissue obtained from rats. The GlyT inhibitors did not alter G-protein basal activity levels (100%) alone or in combination in contrast to DAMGO, a µ-opioid receptor full agonist (140.3% ± 1.58% vs. 100%, DAMGO: *n* = 4, basal: *n* = 6, DF = 42, *p* < 0.001, one-way ANOVA, Dunnett’s post-hoc test; Figure 7). Thus, the applied concentrations of the tested GlyT inhibitors or their combination do not activate G_i/o_ type G-proteins (Figure 7).

## 3. Discussion

The urgent clinical need for effective NP medications with better tolerated side effects leaves many pain researchers seeking for new directions to develop novel analgesics. GlyTs have been identified as promising targets for developing novel drugs that may have future clinical relevance in the management of neuropathic pain [5,15].

The key results of the current study are that the combination of GlyT-1, NFPS (1 mg/kg, s.c.) and GlyT-2 (Org-25543, 2 mg/kg, s.c.) inhibitors produces antiallodynic action in rats with pSNL following acute systemic administration. Each component of the combination namely GlyT-1 inhibitor, NFPS or GlyT-2 inhibitor, Org-25543 failed to show an acute antiallodynic effect when administered alone. Next, NFPS alone in the dose applied in the combination (1 mg/kg, s.c.) but not Org-25543 (2 mg/kg, s.c.) produced significant antiallodynic effect following chronic administration. Both NFPS and Org-25543 only in higher dose (4 mg/kg) produced antiallodynic effect after acute administration. In addition, no motor dysfunction was observed in rats treated with the applied combination or 4 mg/kg dose of subcutaneous NFPS/Org-25543 alone. Direct measurements indicate that the amount of glycine has significantly increased in the CSF of rats treated with the applied combination of GlyTs inhibitors. 

Peripheral sensory fiber (Aβ, Aδ or C) injury contributes to the development of NP conditions. Aδ and C primary sensory afferent fibers transmit pain from the site of injury into the spinal dorsal horn, namely rexed laminae (I-II), whereas Aβ fibers convey non-noxious stimuli to the deeper layer of rexed laminae (II-III) [4,5,16]. Spinal dorsal horn hosts a neural network that has crucial role in the pain transmission toward the brain [4,16]. These neural networks are operated by many inhibitory (GABA and glycine) and excitatory transmitters (glutamate). Convincing evidence has been reported on excitatory and inhibitory neurotransmission imbalance under NP conditions in the spinal dorsal horn [17]. In addition, in NP hypofunction of glycinergic interneurons located in laminae II and III of the spinal dorsal horn has been documented by many research groups [5,18,19,20]. Glycine transporters are widely distributed in CNS including the spinal cord. GlyT-1 is largely distributed in glial cells [21] and has been reported to produce dual effects namely reducing glycine concentration near N-methyl-D-aspartate (NMDA) receptors and participating in inhibitory glycinergic neurotransmission. On the other hand, the spinal GlyT-2 distribution is largely restricted to glycinergic neurons where it transports glycine from the synaptic cleft back to the presynaptic terminals and assures the refilling of vesicles with glycine in the nerve endings of glycinergic interneurons in the dorsal horn laminae II and III [8,21,22]. Physiologically, GlyT-2 works to remove glycine from the synaptic cleft and thereby decreases its concentration, so it is unable to activate glycinergic receptors [10].

In the present work, first we did test the antiallodynic effect of subcutaneous NFPS, GlyT-1 inhibitor, in a dose (1 mg/kg) avoiding motor dysfunction. Indeed, NFPS failed to show acute antiallodynic action, but it produced antiallodynic effect upon chronic treatment. Armbruster and coworkers showed that bitopertin, a GlyT-1 inhibitor, did produce antiallodynic effect following acute treatment in mice [23]. They have also proved that long-term treatment produces antiallodynic effect without noticeable side effects. The same group also reported on the acute antiallodynic effect of bitopertin (1 mg/kg) in CCI-evoked neuropathy in rats after subcutaneous or oral administration [23].

Further, in our present study, in a higher dose (4 mg/kg) NFPS could produce acute antiallodynic effect. In respect to the side effects, previous studies have shown that NFPS in doses of 10–30 mg/kg administered orally or intraperitoneally to mice or rats induces respiratory depression [9,24]. In our study, NFPS in a 4 mg/kg dose avoids producing motor dysfunction assessed by rotarod test or discernible behavioral side-effects related to respiratory depression, however, this dose is only about half of which was previously reported to induce these side effects.

In respect to the GlyT-2 inhibition, acute blockade of this transporter by the irreversible and selective GlyT-2 inhibitor Org-25543, also resulted in the antiallodynic effect in our present work. The applied doses avoided causing motor dysfunction or any discernible behavioral side-effects related to respiratory depression. However, Org-25543 in a dose of 20 mg/kg has been reported to cause a deleterious toxic effect such as spasms, convulsions and death in rodents [14]. On the other hand, in the present work the acute antiallodynic effect of Org-25543 was achieved following subcutaneous administration of 4 mg/kg, which is accounted as one-fifth the dose used by Mingorance-Le Meur and coworkers [14]. 

The current combination contains one tenth of the dose of NFPS and Org-25543 that causes serious side effects, meaning that the strategy’s treatment applied here did produce analgesia with a better safety margin.

Our results to amelioration of chronic NP are in accordance with other studies indicating that GlyT-1 or 2 inhibitors attenuate NP in rodents [6,7,23,25,26]. As mentioned above, GlyT-1 and 2 play crucial roles in the regulation of glycinergic inhibitory neurotransmission, and drugs that affect these transporters are able to affect systems operated by glycine. Next, our hypothesis is that attenuating the transport capacity of both glycine transporters by applying a combination of small inhibitor doses might increase glycine content high enough to produce analgesia. In this regards the combination was relevant, because the combination of NFPS and Org-25543 in subanalgesic doses (1 and 2 mg/kg, respectively) produced an acute antiallodynic effect. In order to explain whether the measured antiallodynic effect is related to an increase in glycine content in the CSF, we analyzed CSF samples using capillary electrophoresis. Applying this method, we could determine rat CSF glycine content, which was close to results reported by another group [27].

Our results showed a significant elevation in CSF glycine content following the acute treatment with the subanalgesic doses of the GlyT-1 and GlyT-2 inhibitors’ combination. However, the subanalgesic doses of each inhibitor failed to produce significant increase in glycine content in the CSF, but a tendency of this increase was noticed. Data on increase in the glycine level in CSF upon treatments of rodents with selective GlyT-1 or GlyT-2 inhibitors have been published by other research groups [28,29,30]. However, to the best of our knowledge, there are no data available on the effects of the GlyT-1 and GlyT-2 inhibitors’ combination on the pain evoked by nerve damage or on the glycine level in the CSF. Many studies have proved that both GlyT-2 and a glial GlyT-1 play an indispensable role in inhibitory glycinergic neurotransmission [5,21,31,32].

Based on our present results and literature data [31], we can hypothesize that upon sciatic nerve injury, increase in glutamate release from central nerve ending occurs. This glutamate increase is more likely to cause the following: internalization of NR1/NR2A NMDA receptors located on glycinergic interneurons as a consequence of massive stimulation. Additionally, glutamate released in excess may spillover into the extrasynaptic place, where it does stimulate non-synaptic NR1/NR2B NMDA receptors. The stimulation of NR1/NR2B NMDA receptor also requires the coagonist glycine released from activated astrocytes. In the presence of adequate concentration of GlyT-1 inhibitor (NFPS), the glycine concentration further increases in the vicinity of the NR1/NR2B NMDA receptor, which in turn may cause internalization of the receptor. On the other hand the GlyT-2 inhibitor, Org-25543 inhibits the reuptake of glycine into the inhibitory glycinergic interneurons synapsing to projection neurons that pass pain sensation toward the brain. The inhibition of the glycine reuptake results in elevation glycine concentration. Thus, simultaneous administration of NFPS and Org-25543 in subanalgesic doses raises glycine concentration in the CSF sufficiently to restore the physiological operation of neuronal circuits in the spinal dorsal horn. Inhibition of this system by concurrent use of GlyT-1 and GlyT-2 inhibitors can interfere with reuptake of glycine and so increase its level at both synaptic and extrasynaptic spaces. The hypothesis is supported by the finding that intrathecally administered strychnine or knockdown of GlyRα3 abolished the analgesic effects of the applied GlyT-1 inhibitor without the potentiation of pain sensation [10,31,33,34,35,36].

Electrophysiological and immunohistochemical evidence shows a stronger accumulation of glycine in neurons expressing GlyT-2 [37,38]. Under present experimental conditions, no difference was detected between the glycine content of tissues obtained from neuropathic and control rats neither between ipsi- and contralateral spinal L4-L6 region of rats underlying pSNL. The same tendency was extended to L-glutamate. Furthermore, L-glutamate was significantly higher in CSF of NP rats compared to control rats. On the other hand, only a smaller, non-significant increase in the glycine concentration of CSF was detected. These results indicate that under NP condition evoked by pSNL, imbalance of inhibitory and excitatory transmission occurs in favor for the latter. Administration of GlyT inhibitors in combination may act by restoring the balance between the inhibitory and excitatory transmission in CSF by further increasing glycine levels.

Combination therapy is a generally applied method to enhance the analgesic effect of drugs acting with different mechanisms of action (e.g., opioids coadministered with non-opioids, e.g., NMDA antagonists) [13]. Here we reported that acute antiallodynic effect was not achieved when rats received either NFPS or Org-25543 in subanalgesic doses. However, the combination of these subanalgesic doses of NFPS and Org-25543 produced an acute antiallodynic effect (Figure 4). When the GlyT inhibitors administered separately in a small dose as mentioned above failed to significantly increase the CSF glycine level. In respect to NFPS, it inhibits the transport of glycine into glial cells nearby the synaptic clefts separating glycinergic interneuron terminals from projection neurons, but the concentration of glycine is not enough to stimulate GlyRα3. GlyT-2 inhibitor, Org-25543 in subanalgesic dose inhibits the reuptake of glycine into the inhibitory glycinergic interneurons synapsing with projection neurons that pass pain sensation toward the brain, but again the amount of glycine is not sufficient to stimulate the glycinergic receptors (GlyRα3) located on projection neurons. Simultaneous administration of NFPS and Org-25543 in subanalgesic doses raises the glycine concentration in the CSF sufficiently to stimulate GlyRα3 (for review see [5,35]). 

Finally, to exclude the direct effect of GlyT-1 inhibitor, NFPS or of the GlyT-2 inhibitor, Org-25543 on G-protein coupling receptors, [^35^S]GTPγS binding assay was applied. Under the present experimental conditions, NFPS and Org-25543 did not activate G-protein coupled receptors, which study, to the best of our knowledge, is the first in its nature related to the GlyT-2 inhibitor [39]. Previously, GlyT-1 inhibitors, NFPS and Org-24461 were found to possess negligible affinity for receptors coupled to G-proteins [39]. In the present study, beside NFPS we also tested Org-25543, a GlyT-2 inhibitor to establish whether or not Org-25543 alone or in combination initiates any Gi protein-mediated signals, which were not investigated in our previous work.

Indeed, the present work opens a new avenue to decrease the unwanted effects of GlyT inhibitors by combining subeffective doses. At the present, NP management, particularly the acute conditions faces obstacles due to the ineffectiveness or side effects of currently available treatments. Such a strategy might be of future clinical relevance at least to subside the acute pain until the onset of the analgesic effect of the current medications is reached. A raised concern is that the combination contains components irreversibly blocking GlyT-1 and 2 [14]. In this regard, we also assessed the possible side effects of the combination of the submaximal analgesic doses of NFPS + Org-25543 and higher doses for both GlyT inhibitors on motor function. These results indicate that the applied doses are devoid of motor dysfunction or any discernible behavioral side-effects under the present circumstances. Our work for the first time shows that the combination of submaximal doses of GlyT inhibitors produces antiallodynic action and devoids motor effects under the present experimental conditions. However, it is important to be aware of the possible toxicity that might be stemmed from the irreversible blockade characteristics of applied compounds. Hopefully, this concern might be circumvented by the combination of reversible GlyT-1 and 2 inhibitors.

## 4. Materials and Methods 

### 4.1. Animals

Male Wistar rats of 100–150 g were subjected to pSNL and male Wistar rats of 170–250 g were used for rotarod test in order to match the weights of operated animals on the experimental days (two weeks after operation). Animals were obtained from Toxi-Coop Zrt. (Budapest, Hungary) and were kept in standard cages in numbers of 4 or 5 animals/cage depending on their weight in a room of 20 ± 2 °C temperature with 12 h/12 h light/dark cycle, in the local animal house of Semmelweis University, Department of Pharmacology and Pharmacotherapy (Budapest, Hungary). Water and standard food were available ad libitum. For DPA measurements altogether 79, for rotarod 51 and for capillary electrophoresis analysis 38 animals were used. 

For G-protein assays both male and female (50–50%, *n* = 4–4) Wistar rats (250–300 g,) were used for membrane preparations obtained from the local animal house of the Biological Research Centre, Hungarian Academy of Sciences (Szeged, Hungary). All the animals were kept at a temperature-controlled room (21–24 °C) under a 12 h/12 h light and dark cycle and were provided with water and food ad libitum.

All housing and experiments were performed in accordance with the European Communities Council Directives (2010/63/EU), the Hungarian Act for the Protection of Animals in Research (XXVIII.tv. 32.§) and local animal care committee (PEI/001/276-4/2013 and PE/EA/619-8/2018). Experimenters put their best efforts to minimize the number of animals and their suffering.

### 4.2. Chemicals

NFPS (*N*-[3-([1,1-Biphenyl]-4-yloxy)-3-(4-fluorophenyl)propyl]-*N*-methylglycine) and Org-25543 (*N*-[[1-(dimethylamino)cyclopentyl]methyl]-3,5-dimethoxy-4-(phenylmethoxy)benzamide hydrochloride) were purchased from Bio-Techne R&D System Kft (Budapest, Hungary). 

For the capillary electrophoresis analysis glycine, L-glutamate, L-cysteic acid, HEPES, acetonitrile and boric acid were purchased from Sigma-Aldrich (St. Louis, MO, USA). 4-Fluoro-7-nitrobenzofurazan (NBD-F) was obtained from the Tokyo Chemical Industry (Tokyo, Japan) and hydroxypropylamino-β-cyclodextrin was provided by Cyclolab Ltd. (Budapest, Hungary). Ultrapure water from MilliQ Direct 8 water purification system (Merck Millipore, Billerica, MA, USA) was used for all experiments. Morphine hydrochloride was obtained from (Alkaloida-ICN, Tiszavasvári, Hungary).

For [^35^S]GTPγS binding assay DMSO, Tris-HCl, EGTA, NaCl, MgCl_2_ × 6H_2_O, GDP and the GTP analog GTPγS were purchased from Sigma-Aldrich (Budapest, Hungary). The highly selective µ-opioid receptor (MOR) agonist enkephalin analog, Tyr-D-Ala-Gly-(NMe)Phe-Gly-ol (DAMGO) was obtained from Bachem Holding AG (Budapest, Hungary). The radiolabelled GTP analog, [^35^S]GTPγS (specific activity: 1250 Ci/mmol, Lot: 0119) and the UltimaGoldTM MV aqueous scintillation cocktail was purchased from PerkinElmer (handled by Per-Form Hungaria Kft, Budapest, Hungary). NFPS and Org-25543 were dissolved in 20% DMSO, while DAMGO was dissolved in ultrapure distilled water for stock solution as indicated on their product sheets. These solutions were further diluted with ultrapure distilled water before administration.

All compounds were stored and handled as described in the product information sheets.

### 4.3. Experimental Protocol

A schematic summary of the applied experimental protocols in this study is presented in Figure 8. First, baseline measurements were performed with DPA (dynamic plantar esthesiometer, see Section 4.5) to assess the pain thresholds before the operation. Afterwards, animals were operated with pSNL (see Section 4.4). On the 14th day after operation, mechanical allodynia was assessed by DPA. Then compounds or vehicles were administered and mechanical allodynia again 30, 60 and 180 min after treatments was determined. These procedures were followed when we determined the acute effect of the test compounds. In experiments designed to assess the chronic effect of the test compounds, animals were further treated once a daily on the 15th, 16th and 17th day, at which day DPA measurements were carried out again similar to the 14th day.

### 4.4. Partial Sciatic Nerve Ligation (pSNL)

pSNL was applied for the induction of mononeuropathic pain in rats, as described before, based on the Seltzer method [40,41]. Briefly, after anesthesia with intraperitoneal 60 mg/kg pentobarbital (in a 2.5 mL/kg volume) the animals were put on a pillow of 30 °C. Under aseptic conditions the operator carefully exposed the sciatic nerve of right hind paw without any muscle damage at the thigh-high level. Then, the nerve was tightly ligated with an 8–0 silicon-treated silk suture in a way that the dorsal 1/3–1/2 of the nerve thickness was trapped in the ligature. The wound was closed with 2 stiches. Sham-operated rats (with the nerve exposed without ligation) were used as controls. 

### 4.5. Assessment of Mechanical Allodynia 

Mechanical allodynia (main symptom of neuropathic pain) was assessed by DPA (dynamic plantar esthesiometer 37450; Ugo Basil, Italy) as described before [40,42]. DPA measures the animals’ paw withdrawal thresholds (PWTs) in grams. PWT values were measured after 5 min of habituation in the cage in the case of each measurement. The DPA equipment raises a metal filament of 0.5 mm diameter to the right and left hind paws alternately, with a force rising from 1 to 50 g (cut-off) based on the manufacturer’s guidelines. The PWT was measured three times on each paw and the average of the measurements was used for further analyses. For each animal, a 20% decrease in the average PWT value of the operated (right) paw compared to the unoperated (left) paw was considered as the development of allodynia [42,43]. Sham-operated animals were used as controls (see Figure A1 in Appendix A). Measurements were carried out as described in Section 4.3 or Figure 8.

### 4.6. Treatment of Animals

NFPS (1 and 2 mg/kg, s.c.) or Org-25543 (2 and 4 mg/kg, s.c.) were investigated, after acute treatment on the day 14 after pSNL. The applied doses and the routes of administration were chosen based on the previous studies that have examined the pharmacodynamic and pharmacokinetic profiles of NFPS or Org-25543 [5,10,14,26,33,39,44]. NFPS in 1 mg/kg and Org-25543 in 2 mg/kg were investigated for their chronic effect after 4 days of treatment (once a day for 4 days). In the pilot study, chronic effect of NFPS was investigated on the 6th day also. As data obtained for 4- or 6-day treatments were identical, thus in the further experiments 4-day treatment protocol was adopted. In each treatment 3–9 animals per group were applied.

### 4.7. Motor Function Test

The effect of test compounds on motor coordination of animals was evaluated by the rotarod test (Rat Rotarod, Model 7750; Ugo Basile, Gemonio, Italy). One day before the experiment animals were trained to stay on the rotating rod of the instrument. The speed of the instrument was set to 16 rpm and the cut off time was maximized in 180 s. On the next day, following subcutaneous treatment with NFPS, Org-25543 and their combination or vehicle (control) animals were tested at the time of peak effect of test compounds. The time-latencies were noted in seconds (fall-off time). Morphine (6.4 mg/kg, s.c.) was tested acutely at its peak-effect time (60 min), as a positive control.

### 4.8. Capillary Electrophoresis Analysis of Glycine and Glutamate Content

Glycine and glutamate content of spinal cord and cerebrospinal fluid (CSF) samples was measured by capillary electrophoresis-laser induced fluorescence method developed in our laboratory [45] with some modifications.

Neuropathic and sham operated control rats were sacrificed 14 days after pSNL operation. CSF samples were obtained by cisterna magna puncture and centrifuged at 2000× *g*, 4 °C for 10 min. Lumbar 4–6 segment of the spinal cord was removed. Samples were frozen immediately and stored at −80 °C until further processing. On the day of experiments spinal cord samples were disrupted by ultrasonic homogenizer for 20 s in acetonitrile-distilled water solution (2:1 *v*/*v*; 20 μL/mg tissue). Samples were then centrifuged at 20,000× *g* for 10 min at 4 °C in order to remove precipitated proteins. CSF samples were deproteinized by mixing with 2 volumes of pure acetonitrile and centrifuged at 20,000× *g* for 10 min at 4 °C. Supernatants from each sample type were collected, diluted five times with acetonitrile-distilled water solution (2:1; *v*/*v*) and subjected to derivatization with NBD-F (1 mg/mL final concentration) in 20 mM borate buffer pH 8.5 for 20 min at 65 °C. 5 µM L-cysteic acid was used as internal standard.

Derivatized samples were analyzed by a P/ACE MDQ Plus capillary electrophoresis system coupled with laser induced fluorescence detector equipped with a laser source of excitation and emission wavelengths of 488 and 520 nm, respectively (SCIEX, Framingham, MA, USA). Separations were carried out in polyacrylamide coated fused silica capillaries (i.d.: 75 µm, effective/total length: 50/60 cm) using 50 mM HEPES buffer pH 7.0 containing 6 mM hydroxypropylamino-β-cyclodextrin at 15 °C by applying −30 kV constant voltage.

### 4.9. G-protein Activity Assay

Rat spinal cord membrane preparation: Rats were decapitated and their spinal cords were quickly removed. The spinal cords were prepared for membrane preparation as previously described for the [^35^S]GTPγS binding experiments [46]. Briefly, the spinal cords were homogenized in ice-cold TEM buffer composed of 50 mM Tris-HCl, 1 mM EGTA, 3 mM MgCl_2_, and 100 mM NaCl with a Teflon-glass homogenizer. The homogenate was centrifuged at 18,000 rpm for 20 min at 4 °C, the resulting supernatant was discarded and the pellet was further incubated at 37 °C for 30 min in a shaking water-bath. Then, centrifugation was repeated as described above. The final pellet was suspended in ice-cold TEM pH 7.4 (50 mM Tris-HCl, 1 mM EGTA, 3 mM MgCl_2_, and 100 mM NaCl) buffer and stored at -80 °C. The protein content of the membrane preparation was determined by the method of Bradford, BSA being used as a standard [47].

Functional [^35^S]GTPγS binding assay: In [^35^S]GTPγS binding experiments we measured the GDP → GTP exchange of the Gαi/o protein in the presence of a given ligand. The nucleotide exchange was monitored by a radioactive, non-hydrolysable GTP analogue, [^35^S]GTPγS. The functional [^35^S]GTPγS binding experiments were performed as previously described [48,49], with modifications. Briefly, the membrane homogenates were incubated at 30 °C for 60 min in TEM buffer (pH 7.4, 50 mM Tris-HCl, 1 mM EGTA, 3 mM MgCl_2_, and 100 mM NaCl), containing 20 MBq/0.05 mL [^35^S]GTPγS (0.05 nM) and 0.1–10 µM concentrations of the GlyT inhibitors (alone or in combination) and DAMGO. The experiments were performed in the presence of excess GDP (30 µM) in a final volume of 1 mL. Total binding was measured in the absence of test compounds, non-specific binding was determined in the presence of 10 µM unlabeled GTPγS and subtracted from total binding. The difference represents basal activity. The reaction was terminated by rapid filtration under vacuum (Brandel M24R Cell Harvester, Gathersburg, MD, USA), and washed three times with 5 mL ice-cold 50 mM Tris-HCl (pH 7.4) buffer through Whatman GF/B glass fibers. The radioactivity of the filters was detected in UltimaGold^TM^ MV aqueous scintillation cocktail with Packard Tricarb 2300TR liquid scintillation counter (Per-Form Kft, Budapest, Hungary). [^35^S]GTPγS binding experiments were performed in triplicates and repeated at least three times.

### 4.10. Statistical Analysis

All data were presented as mean ± SEM. Data were analyzed by two-way ANOVA in case of DPA measurements when the effect of two factors, namely treatment and time was to be determined. In other cases (rotarod test, CSF and spinal amino acid content and G-protein activation assay), one-way ANOVA was applied to analyze the effect of one single factor. For comparison of multiple groups Newman–Keuls, Holm–Sidak and Dunnett’s post-hoc tests were used for in vivo studies, amino acid level determination and G-protein binding assay, respectively. In case of DPA measurements, vehicle-treated animals were considered as a control group. The differences were considered significant if *p* < 0.05. Data analysis was carried out by the professional statistical software GraphPad Prism 6.0 (GraphPad Software Inc., San Diego, CA, USA).

## 5. Conclusions

Summarizing, treatment with GlyT-1 or GlyT-2 inhibitors at higher doses produced acute antiallodynic effect in rats with mononeuropathic pain. Combination of GlyT-1 with GlyT-2 inhibitors did show acute antiallodynic effect in doses that separately failed to produce analgesia. Either the doses applied in the combination or the combination itself did not cause motor dysfunction or any discernable behaviors. The observed analgesic effect of the combination was parallel with increased glycine content of the cerebrospinal fluid. The test compounds or their combination failed to show activity in the G-protein coupling activity assay, indicating no G-protein coupled mediated effects at the spinal level. Based on these results the combination strategy applied in this study might offer safe and effective therapies for future acute NP management.

## Figures and Tables

**Figure 1 ijms-22-02479-f001:**
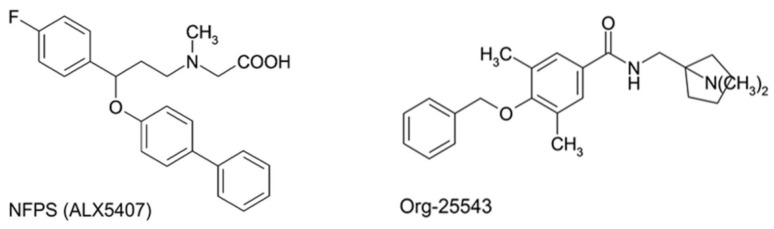
Structural formula of GlyT-1 inhibitor, NFPS (N-[3-([1,1-Biphenyl]-4-yloxy)-3-(4-fluorophenyl)propyl]-N-methylglycine) also known as (ALX5407) for its isomeric form, and GlyT-2 inhibitor, Org-25543 (N-[[1-(dimethylamino)cyclopentyl]methyl]-3,5-dimethoxy-4-(phenylmethoxy)benzamide hydrochloride).

**Figure 2 ijms-22-02479-f002:**
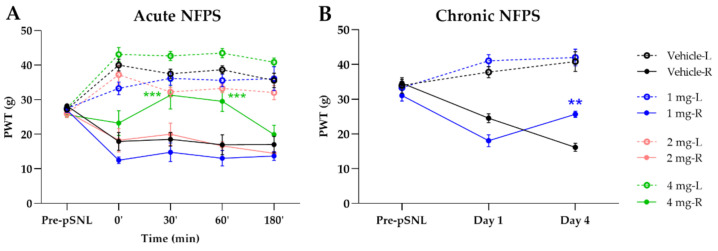
The antiallodynic effect of NFPS following acute (**A**) and chronic treatment (**B**). Graphs show the means of PWT ± S.E.M. in grams, of animals’ left (healthy, L) and right (operated, R) paw before (pre-partial sciatic nerve ligation (pSNL)) and after the pSNL in the indicated time points or days and treatment groups. Graphs for sham operated groups are excluded for better visual clarity, however they are presented in Figure A1. Day 1 and 4 represent the days of treatment, where the 60 min of PWT ± S.E.M. values are indicated on the current day. asterisk: indicates the significant differences compared to the vehicle group (two-way ANOVA, Newman–Keuls post-hoc test; ***: *p* < 0.001; **: *p* < 0.01). In each treatment group 4–5 animals were used.

**Figure 3 ijms-22-02479-f003:**
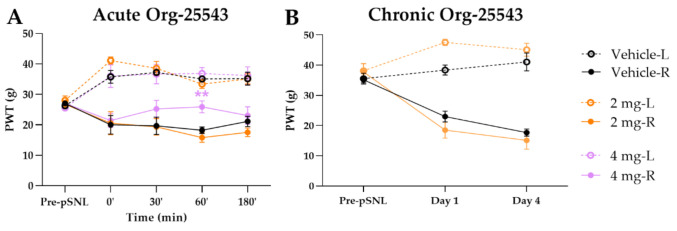
The antiallodynic effect of Org-25543 following acute (**A**) and chronic treatment (**B**). Graphs show the means of PWT ± S.E.M. in grams, of animals’ left (healthy, L) and right (operated, R) paw before (pre-pSNL) and after the pSNL in the indicated time points or days and treatment groups. Graphs for sham operated groups are excluded for better visual clarity, however they are presented in Figure A1. Day 1 and 4 represent the days of treatment, where the 60 min of PWT ± S.E.M. values are indicated on the current day. asterisk: indicates the significant differences compared to vehicle group (two-way ANOVA, Newman–Keuls post-hoc test; **: *p* < 0.01). In each treatment group 4–6 animals were used.

**Figure 4 ijms-22-02479-f004:**
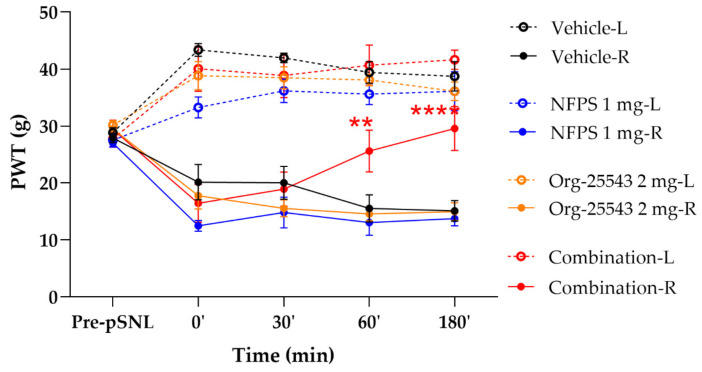
The antiallodynic effect of the combination of NFPS 1 mg/kg s.c. and Org-25543 2 mg/kg following acute treatment. Graphs show the means of PWT ± S.E.M. in grams, of animals’ left (healthy, L) and right (operated, R) paw before (pre-pSNL) and after pSNL in the indicated time points and treatment groups. Graphs for sham operated groups are excluded for better visual clarity, however they are presented in Figure A1. asterisk: indicates the significant differences compared to vehicle group (two-way ANOVA, Newman–Keuls post-hoc test; ****: *p* < 0.001; **: *p* < 0.01). In each treatment group 4–5 animals were used.

**Figure 5 ijms-22-02479-f005:**
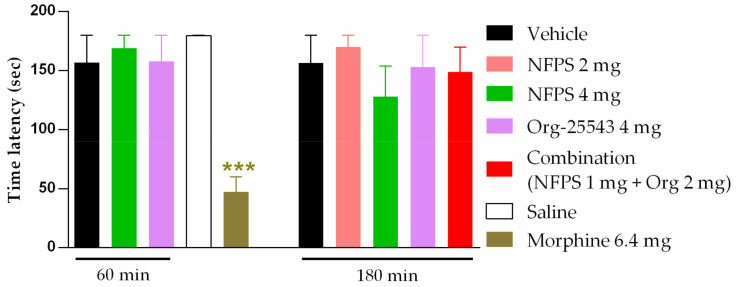
Effect of systemic administration of analgesic doses of NFPS (2 or 4 mg/kg s.c.) and Org-25543 (4 mg/kg s.c.), and combination of their sub-analgesic doses (NFPS 1 mg/kg + Org-25543 2 mg/kg s.c.). Columns represent the time latency of the animals in sec ± S.E.M. at 60 and 180 min for GlyT inhibitors and at 30 min for morphine post-treatment in rotarod test. asterisk: indicates the significant differences compared to saline group (one-way ANOVA, Newman–Keuls post-hoc test; ***: *p* < 0.001). In each treatment group 4–7 animals were used.

**Figure 6 ijms-22-02479-f006:**
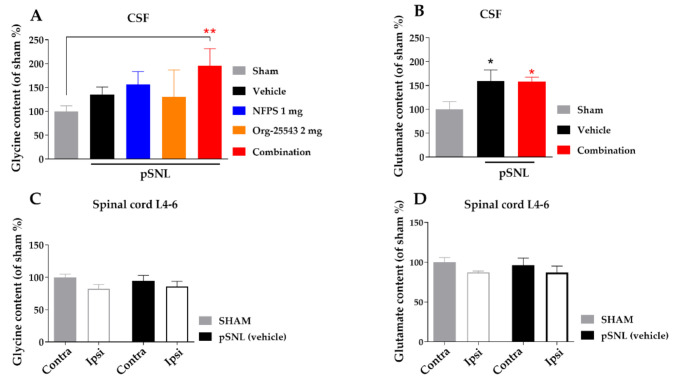
The glycine and L-glutamate content in CSF (**A**,**B**) and in spinal cord L-4-6 ipsi- and contralateral side (**C**,**D**) obtained from sham and rats underwent pSNL after 14 days. Columns represent the given amino acid content S.E.M in %. in the indicated groups. Concentration levels were normalized to sham and shown in percentage. asterisk: marks the significant differences compared to sham group (one-way ANOVA, Holm–Sidak post-hoc test; **: *p* < 0.01, *: *p* < 0.05). In each treatment group 3–11 animals were used.

**Figure 7 ijms-22-02479-f007:**
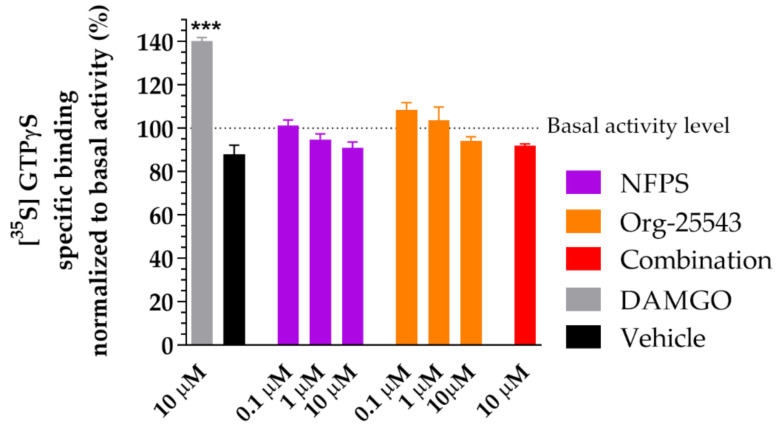
The impact of glycine transporter inhibitors NFPS and Org-25543 compared to DAMGO, a µ-opioid receptor agonist as reference compound on G-protein activity in the rat spinal cord membranes. Compounds were added in the indicated concentrations. For comparison, vehicle (DMSO 20%) was also measured. Columns represent the means ± S.E.M. of the specific binding of [^35^S]GTPγS normalized to basal activity (100%). asterisk: indicates the significant alteration compared to basal activity (one-way ANOVA followed by a Dunnett′s multiple comparison test ***: *p* < 0.001). All experiments were repeated at least 4 times, using 3 parallels.

**Figure 8 ijms-22-02479-f008:**
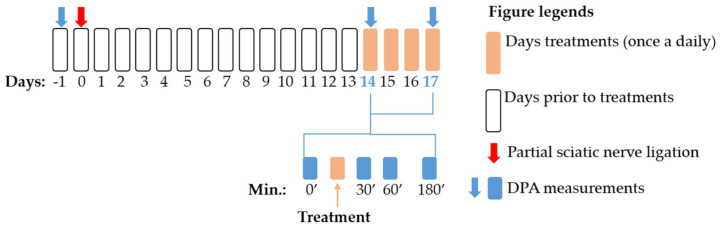
Schematic representation of the experimental protocol applied in the study. The figure indicates the timeline of the dynamic plantar esthesiometer (DPA) measurements, pSNL and treatment days. The figure also highlights the exact time points for DPA measurements within the treatment days.

## Data Availability

The data that support the findings of this study are available from the corresponding author upon reasonable request.

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
