# Peer review of "Pharmacological Evidence on Augmented Antiallodynia Following Systemic Co-Treatment with GlyT-1 and GlyT-2 Inhibitors in Rat Neuropathic Pain Model"

_ijms, 2021, doi:10.3390/ijms22052479_

Round 1

Reviewer 1 Report

In this article, Amir Mohammadzadeh et al. investigated the analgesic effect of two glycine transporter inhibitors, NFPS and Org-25543, in rat neuropathic pain model – partial sciatic nerve ligation. The main findings of this study were that co-treatment with low, sub-analgesic doses of these inhibitors given subcutaneously produced analgesia and increased a glycine content in the cerebrospinal fluid.

The rationale and study design seem to be quite reasonable and the experiments well done. However, the manuscript has several shortcomings. I suggest the authors clarify or provide more details regarding the following points:

  1. The Discussion section: lines 185 – 187, the effective doses of the compounds should appear here. Please, modify the sentence.
  1. Discussion, lines 303 – 306; The lack of affinity of NFPS and Org-25543 towards main classes of G-protein-coupled receptors (GPCRs) was described earlier by Harsing et al., 2003 (as is cited by the Authors). Moreover, neither glycine receptors nor NMDA receptor belong to the GPCRs as they are ion channels. Thus, the rationale for the G protein activity assay experiment in the current study should be discussed in more depth either in the Discussion or Introduction section.
  1. Methods section: lines 392 – 398. On what basis were the doses of NFPS and Org-25543 inhibitors selected for the experiments? Please, provide readers with some information on this issue.
  2. There is an editorial problem with this manuscript, i.e., with the numbering of subsections, placing general conclusions after methodological sections, etc. It would be very useful if the authors could read a second time the paper to correct carefully the manuscript organization.

Reviewer 2 Report

Main comments:

Materials and methods section has to be corrected and completed to understand causal connection of used methods, enable correct assessment of the quality of performed studies as well as to redo the experiments in the future.

  • There is no given rationale for the choice of route of administration as well as doses of GlyT-1 and GlyT-2 inhibitors used in the presented study for the acute application (NFPS – 1 and 2 mg/kg/s.c.; Org-25543 - 2 and 4 mg/kg/s.c.) and for the chronic treatment (NFPS – 1mg/kg/s.c.) of allodynia in rat model of neuropathic pain. Authors, in discussion section compare obtained results to literature data showing effects of other GlyT inhibitors used in similar doses or to data using much higher dose of Org-25543 (20 mg/kg). There is no, however, clear explanation/ discussion of pharmacokinetic parameters of compounds used in the study. Please complete section 2.6 (Treatment of animals, lines 392-398) with the explanation of the choice of route of administration as well as choice of NFPS and Org-25543 doses.
  • Please provide information which hind leg of rat had partial sciatic nerve ligation in methodological description of this procedure in the section 4.2.4
  • Information about duration of chronic treatment with NFPS is not consistent throughout the manuscript. E.g. line 36 (Abstract) says “three days of treatment”; in line 105 and 120 (Figure 2 and 3 descriptions) is information about 4 days “chronic treatment (day 1 to 4)”; based on scheme in Figure 8 – chronic treatment was for only two days; line 395 (Materials and Methods) says "6 days". Please correct inconsistencies.
  • Schematic summary of experimental protocol presented in Fig. 8 is hard to understand. E. g. it is not clear from Fig. 8. what is the difference between “Day of the experiment (in the morning)” indicated in black and “days of chronic treatments and presented protocol (once daily)” indicated in grey; time schedule is not easy to understand also (minus before 14 looks like hyphen to word “Day”); grey color suggest only 2 days of “chronic treatment”. Please make the scheme presented in Fig 8 self-describing. It would be helpful for readers if you could include in the scheme or describe in the body text, all experimental groups generated using protocol presented in Fig 8 with the information about number of rats per each group.
  • Information about number of animals used to conduct experiments presented in the manuscript is lacking. Please complete necessary information about number of animals in performed experiments in the section 4.2.1.(lines 324-340).
  • Information on the origin a few reagents is missing: MORPHINE used as a positive control in rotarod test; L- GLUTAMATE used to prepare results presented in Figure 6B and 6D. Please complete information about reagents origin in the section 4.2.2.(lines 341-360)
  • Information about statistical analysis performed to obtain data presented in Figs 2- 7 is laconic. Please complete the description of statistical methods in section 2.10 (line 454 – 461) including the explanation of the reasons of different analysis (1 way, 2 way, student’s test) for presented data.
  • Lines 414-416: Did you homogenized or sonicated samples? Please consider if you mean >samples were disrupted by ultrasonic homogenization<
  • Line 432 – did you mean TEM buffer?
  • Line 469: “ no receptorial effects”- did you mean >no related to G protein coupled receptor effects<?

Result presentation in Figures, Figure captions and the body text in Results section have to be improved

  • Please complete description of obtained results in all subsections of section 2 (Results), including description of statistical analysis with the information about means, errors (SE, SD or SEM), in case of parametric tests or median values with quartiles in case of non-parametric tests. Please provide also appropriate report of statistics for your test (t value for t-test, F for ANOVA etc), accompanied with the sample size (non-parametric tests) or degrees of freedom (df; parametric tests).
  • In Figures, Figure Legends or Figure descriptions please provide number of animals per each group.
  • In Figure descriptions please remove redundancy (e.g. explanation of abbreviations which were provided earlier in the manuscript like in line 121: “pSNL (partial sciatic nerve ligation)”; detailed explanation of experimental protocol which were provided in Material and Methods section like in lines 121-122: “operation was 14 days before experimental day 1”; if you give information about rout of treatment like s.c. there is no necessary to give information that treatment were “systemic” (see e.g. line 119))
  • In Figure descriptions please describe in which way the symbols/ bars are expressed (mean/median, SD, SE, etc); E.g. >Data are expressed as a mean values ± SEM<
  • Please correct figure 5 or description for bar “20%DMSO…” Line 143 says value 180±0s for this group while bar indicates higher then stated error bar. Also there is no error bar for “saline…” group – please make sure you provided correct data for publication.
  • In Figure descriptions for p-values give precise information versus which group result is significantly different.

Discussion is rather sparse regarding justification of potential mechanism of therapeutic activity and lower adverse effects of GlyT inhibitors applied in manner presented in manuscript and should be improved.

  • Please discuss more precisely possible mechanisms of GlyT inhibitors responsible for allodynia alleviation. Moreover, it is not clear from the manuscript how GlyT1 and GlyT2 cooperate on cellular/systemic level in pain management. Please elaborate discussion in the content on possible specific cellular mechanisms of anti-pain activity of studied GlyT inhibitors.

Minor comments:

  • The English style/grammar of the manuscript needs corrections. To avoid misunderstanding, my recommendation for authors is to read the revised manuscript carefully before they send it again to Editor to avoid typo, unclear sentences.

Sentences I found unclear e.g.:

Line 57: “…potential vanilloid receptor 1 (TRPV1) receptors.” Did you mean: >… potential vanilloid receptor 1 (TRPV1) agonists<?

Line 74-75: “….adverse effects are phenotype dependent” – not clear which phenotype authors are talking about.

Line 86: “Finally, to exclude the G-protein mediated agonist activity of NFPS…”. Did you mean: >Finally, to exclude agonistic activity of NFPS to G-protein coupled receptors <?

In line 386 should be >analysys<

  • Once entered, the abbreviation should be used consistently. I found that the abbreviations and full names are used in the article by chance (E.g. “NP” abbreviation is introduced in line 50 instead in line 49 when first time is used in main text. Authors often do not use introduced abbreviation NP- see e.g. lines 71, 183, 380; “pSNL” abbreviation is introduced several times in the body text but not when first time is used in line 63; “PWT” is used for description of Fig 2 (line 103) while is explained in Figure 8 description (lines 365-366).

There is mistake in the abbreviation introduced in the Abstract: “GlyTs” or rather “GlyT” should be moved from line 30 to line 29 before word “Inhibitors”.

Reviewer 3 Report

It has been well known for more than a decade that GlyT- inhibition could reduce the neuropathic pain in animal model. This manuscript aims to exam the effect of 2 GlyT inhibitors simultaneously for neuropathic pain treatment. Since GlyT1 and T2 are expressed at different cell, the combined effect of 2 drugs is anticipated. However, author did not show the glycine content changed in the spinal cord L4-6 with drug application. There is no mechanistic insight about the co-application of GlyT1 and T2 inhibitor. The overall significant of this manuscript is lacking.

There are some concerns about the manuscript.

  1. The n number in figure 2A is 4-14. This high variation may cause false statistical analysis.
  2. There should be additional time point to test the side effect test for motor function. For this manuscript, the drug along produced best effect at 60 min after treatment. Since 30 min is used for morphine. Therefore, the side effect should also include 60 min time point or even 30 min.

Some minor suggestion for the manuscript.

  1. The graph is hard to read. For Figure 2, the color and symbols are hard to see. For example, is NFPS 1mg/kg result missing in the 2A? or is it overlapping with another line? The significant labeling is unclear. What is significant different of NFPS(4mg/kg)-R compared to?
  2. In title, achieved is redundant.
  3. Figure 8 should be the first figure of the manuscript to help reader understand the design of experiment.
  4. Information is missing in figure 8. Chronic treatment of Org-25543 is missing.

Round 2

Reviewer 1 Report

The Authors have been able to correctly revise my previous comments on the manuscript. I have no additional comments for the Authors.

Reviewer 2 Report

The authors addressed all my critical remarks and applied necessary corrections into the current version of the manuscript, which significantly improved its scientific quality.

Reviewer 3 Report

I have no further comments